# Vasculitis Secondary to Pulmonary Bacterial Infection: A Case Report

**DOI:** 10.3390/diagnostics12040772

**Published:** 2022-03-22

**Authors:** Wangji Zhou, Wei Ye, Juhong Shi, Sanxi Ai, Xinlun Tian

**Affiliations:** 1Department of Pulmonary and Critical Care Medicine, State Key Laboratory of Complex Severe and Rare Diseases, Peking Union Medical College Hospital, Chinese Academy of Medical Sciences, Peking Union Medical College, Beijing 100730, China; zhouwj28@foxmail.com (W.Z.); juhong_shi@hotmail.com (J.S.); 2Department of Nephrology, Peking Union Medical College Hospital, Chinese Academy of Medical Sciences, Peking Union Medical College, Beijing 100730, China; pumch-yw@163.com

**Keywords:** vasculitis, infection, infection-related glomerulonephritis

## Abstract

Vasculitides are a heterogeneous group of primary disorders which may occur secondary to a variety of conditions. Among them, vasculitis caused by bacterial infection is rare. Here, we present images of CT scans and histology from a 22-year-old young Chinese man with vasculitis secondary to bacterial infection, which is a difficult disease to diagnose. This patient had been diagnosed with antineutrophil-cytoplasmic-antibody-negative vasculitis with pulmonary and renal involvement and was treated with glucocorticoids combined with immunosuppressive agents. However, during his follow up we found that multiple patchy shadows and cavities in both lungs and renal lesions had fluctuated, and the improvement of chest imaging was always related to antibiotic treatment. In addition, renal histology showed capillary loop necrosis and extensive crescent formation, and electron microscopy revealed scattered subepithelial hump-like deposits, which favored the diagnosis of infection over idiopathic vasculitis. Therefore, vasculitis secondary to infection was confirmed. The subsequent therapy response supported our diagnosis. This case is important; since vasculitis secondary to infection is uncommon, our case provides a model for the diagnosis of vasculitis secondary to infection.

## 1. Introduction

Vasculitides are a heterogeneous group of disorders characterized pathologically by vascular inflammation, damage and tissue necrosis [1]. These disorders may be primary or occur secondary to a variety of conditions, such as infections, tumors or diffuse connective tissue diseases [2]. Infectious agents such as bacteria can lead to secondary vasculitis, usually not associated with ANCA, and affect vessels of various calibers [3].

Similar to other primary systemic vasculitides, vasculitis secondary to infections can also involve multiple organs, including the lungs, kidneys, nervous system and skin [4]. Here, we reported a 22-year-old man with productive cough and multiple patches and cavities in the lungs who was ultimately diagnosed with vasculitis secondary to pulmonary bacterial infection.

## 2. Case Presentation

A 22-year-old man visited our hospital in April 2021 for intermittent productive cough with multiple patchy shadows and cavities in both lungs for 3 years. In November 2017, January 2018 and March 2018, the patient was admitted to a local hospital three times due to fever, cough and expectoration. The chest CT showed patchy shadows in both lungs and empiric antibiotic treatment was effective. However, when the patient’s symptoms recurred in July 2018, the chest CT showed patchy shadows in the upper lobes of both lungs and a small cavity in the right lower lobe. In addition, there was no significant improvement after empirical antibiotic therapy. After 3 months of empiric anti-tuberculosis therapy, the lesions progressed. During his first hospitalization in our hospital in December 2018, the patient presented with microhematuria, proteinuria (5.32 g per day), elevated serum creatinine (SCr, 177 µmol/L, baseline 75 µmol/L; normal, 59–104) and decreased complement 4 (C4, 0.091 g/L; normal 0.100–0.400). During this time, the patient developed a purpuric rash on both lower extremities. After investigations including autoantibody testing, bronchoscopic mucosal biopsy and renal biopsy, he was diagnosed with antineutrophil cytoplasmic antibody (ANCA)-negative vasculitis with pulmonary and renal involvement. In the following 2 years, the patient was treated with glucocorticoids combined with immunosuppressive agents (cyclophosphamide, azathioprine and mycophenolate mofetil) as well as intermittent antibiotics. His SCr and C4 levels returned to normal, but his chest imaging findings and urinary protein levels fluctuated. In his past medical history, at the age of 12, the patient had been diagnosed with Henoch–Schönlein purpura, which improved with glucocorticoid treatment. Over the past 3 years, the patient intermittently developed skin purpura. He was an ex-smoker, having quit 3 years prior to presentation. There was no family history of lung disease.

His vital signs were stable. Scattered purpura and obsolete patches were seen on both lower legs, mostly on the ankles and on the extensor side. The rash was of needle tip size and flat on the skin surface. No peripheral lymphadenopathy was noted upon palpation. Lungs were clear to auscultation. No wheezing was noted. The results of the rest of the examination were normal.

Laboratory results showed that routine blood tests and liver function and renal function parameters were normal. The erythrocyte sedimentation rate was 111 mm/h (normal, 0–15), and the C-reactive protein level was 33.81 mg/L (normal, <8.00). Antinuclear antibodies, extractable nuclear antigens and ANCA were negative. Complement 3 (C3), C4 and cryoglobulin were within normal ranges. The serologies of hepatitis B virus, hepatitis C virus, syphilis and HIV were negative. Purified protein derivative, cytomegalovirus DNA and Epstein–Barr virus DNA tests, as well as interferon-gamma release assays, were negative. Pulmonary function tests showed decreased diffusion capacity (percent predicted diffusing capacity of lungs for carbon monoxide, DLco% = 54.4%) without obstructive or restrictive dysfunction. Abdominal and pelvic CT examinations were performed and excluded an underlying malignancy.

Given the poor effect of glucocorticoids and immunosuppressive therapy over the past 2 years, we reviewed the patient’s previous test results. In November 2018, the patient underwent bronchoscopy in our hospital for the first time. Bronchial mucosal pathology suggested significant chronic inflammation of the mucosa, with no definite epithelioid granulomas or caseous necrosis. The pathological results of a renal biopsy in December 2018 suggested endocapillary proliferative (Figure 1). Electron microscopy revealed scattered subepithelial hump-like deposits, suggesting endocapillary proliferative glomerulonephritis.

Changes in chest CT imaging, SCr, 24 h urine protein (24 h-UP) and the treatment of the patient are summarized in Figure 2 and Figure 3. The patient underwent four bronchoalveolar lavage fluid (BALF) cultures in November 2018, April 2019, November 2019 and April 2021, revealing persistent positive *Pseudomonas aeruginosa* and positive benzoxicillin-sensitive *Staphylococcus aureus* twice (November 2018 and November 2019).

On the basis of careful review of chest imaging, SCr, 24 h-UP and the treatment of the patient, we found that the improvement in chest imaging findings was always related to antibiotic treatment, whereas immunosuppressive therapy did not lead to improvement. In April 2019 in particular, in the course of treatment with sufficient glucocorticoids and cyclophosphamide, the decreased 24 h-UP showed a significant rebound.

The key findings that prompted us to revise the diagnosis were those of renal pathology. With light microscopy, endocapillary hypercellularity was observed with extensive crescent formation and segmental necrosis in capillary loops, which suggested necrotizing glomerulonephritis. As common causes of necrotizing glomerulonephritis, anti-glomerular basement membrane disease, lupus nephritis and ANCA-associated vasculitis could be excluded through clinical manifestations and antibody testing results. Immunofluorescence showed C3 deposition in mesangial areas and capillary loops, and the absence of IgA deposition did not support the diagnosis of IgA nephropathy or Henoch–Schönlein purpura nephritis (IgA vasculitis). Finally, combined with scattered subepithelial hump-like deposits on electron microscopy, we considered the final diagnosis of renal damage to be infection-related glomerulonephritis.

Considering that the diagnosis of infection-associated glomerulonephritis was clear, the thin-walled cavity of the lung could not be explained by lung injury directly caused by infection. Therefore, despite the lack of the appropriate lung pathological findings, we believe that the patient’s lung lesions were the pulmonary manifestations of infection-induced vasculitis. The subsequent treatment response confirmed our assumption.

*P. aeruginosa* and *S. aureus* were isolated from the BALF of the patient several times. According to the drug sensitivity test, the patient received oral levofloxacin (0.5 g/d) for two weeks, followed by aerosol inhalation of amikacin sulfate (0.2 g) twice a day. Given that steroids may attenuate the systemic inflammatory response, we continued to maintain glucocorticoids at low doses (oral prednisone, 10 mg/d). After 2 months of treatment, the cough and expectoration of the patient were significantly relieved. A chest CT scan showed that the multiple patches and cavities in the lung were obviously absorbed (Figure 4). The 24 h-UP and renal function parameters remained normal.

## 3. Discussion

We reported a difficult-to-diagnose case of vasculitis secondary to pulmonary bacterial infection. Vasculitides are a heterogeneous group of disorders characterized pathologically by vascular inflammation, damage and tissue necrosis. These disorders may be primary or occur secondary to a variety of conditions, such as infections, tumors or diffuse connective tissue diseases. Among vasculitides, infectious agents can be an important potential cause of vasculitis, usually not associated with ANCA, and affect vessels of various calibers.

The most common pathogens associated with vasculitis are viruses. In addition to viruses, bacteria may also be responsible for vasculitis [5]. Streptococcus or Staphylococcus infection is the prototypical cause of acute postinfectious glomerulonephritis [6]. *P. aeruginosa* infection is associated with destructive lung vasculitis [3].

The direct invasion of blood vessels by pathogens or septic hematogenous embolization plays a role in the pathogenesis of vasculitis secondary to infections [4]. On the other hand, excess antigen leads to increased circulating immune complexes deposited on and within the vessel wall, and the subsequent activation of complement results in vessel wall damage [5]. In addition, autoimmune T- and/or B-cell reactions elicited by microbial antigens may be functional in some cases, possibly because of epitope mimicry or superantigen-induced T-cell activation [3].

Clinical scenarios suggestive of vasculitis include diffuse alveolar hemorrhage, deforming or ulcerating upper airway lesions, cavitary or nodular disease on chest imaging, rapidly progressive glomerulonephritis, mononeuritis multiplex and palpable purpura [7]. However, it is difficult to differentiate vasculitis secondary to infection from primary systemic vasculitis based on their clinical manifestations. The former can similarly involve multiple organs, including the lungs, kidneys, nervous system and skin, as with primary systemic vasculitis [4].

Infection-related glomerulonephritis tends to be more common in school-aged children than in adults. Symptomatic patients most commonly manifest acute nephritic syndrome, with new-onset hematuria and proteinuria, edema, hypertension and reduced renal function [6]. Extrarenal manifestations may include a Henoch–Schönlein purpura-like presentation. Hypocomplementemia is present in 35–80% of adults with postinfectious glomerulonephritis, compared with approximately 90% of children. In most patients, C3 is depressed with or without the depression of C4 and normalizes within 2 months [8].

Both the disease itself and the use of immunosuppressive agents contribute to the higher risk of infection in these patients, making the diagnosis of vasculitis secondary to infections more difficult [9]. We suggest that in the initial evaluation of vasculitis, clinicians should pay attention to the possibility of vasculitis secondary to infection. If the differential diagnosis is confusing, it is also feasible to treat and observe both vasculitis and infection at the same time. Remission following anti-infective therapy or recurrence during active immunosuppressive therapy often supports vasculitis secondary to infections.

For infection-related glomerulonephritis, the following features are helpful in making the diagnosis: (1) clinical or laboratory evidence of infection preceding or at the onset of glomerulonephritis; (2) depressed serum complement; (3) endocapillary proliferative and exudative glomerulonephritis; (4) C3-dominant or codominant glomerular immunofluorescence staining; (5) subepithelial hump-like deposits on electron microscopy [8].

The treatment of vasculitis caused by bacteria mainly depends on antimicrobial agents, and a short course of glucocorticoid and/or immunosuppressive agents may sometimes be necessary [5]. In terms of postinfectious glomerulonephritis, the treatment includes the eradication of infection and management of the complications of nephritis. Most postinfectious glomerulonephritis self-heals after the infection is controlled. However, if there is a combination of necrosis and crescents, steroid therapy is often needed. If the number of crescents is large, combined immunosuppressive therapy is often necessary [8].

## 4. Conclusions

We reported a rare case of vasculitis secondary to pulmonary bacterial infection, and the diagnostic process was extremely difficult. Recognizing an infectious origin of vasculitis is of great importance because treatment strategies differ from those applied to non-infectious forms. Although vasculitis secondary to infection is uncommon, we suggest that in the initial evaluation of vasculitis, clinicians should pay attention to the possibility of vasculitis secondary to infection.

## Figures and Tables

**Figure 1 diagnostics-12-00772-f001:**
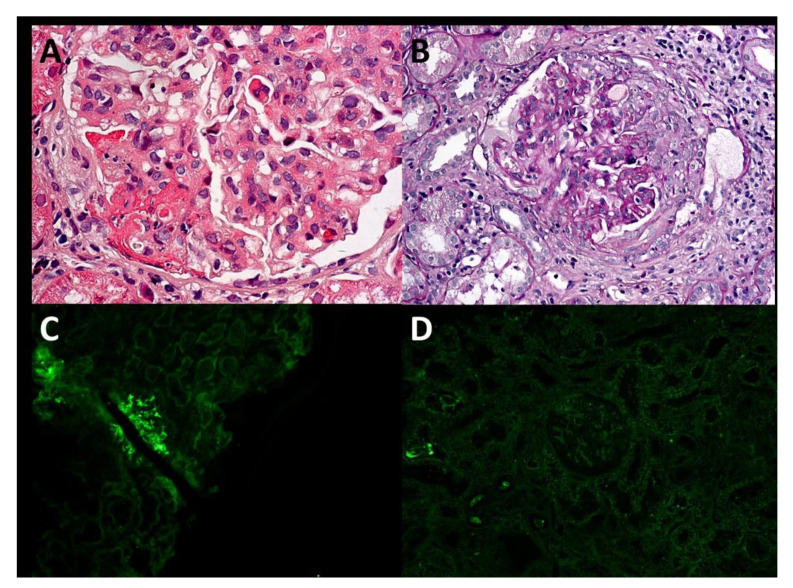
Renal histopathology. Light microscopy (**A**,**B**), immunofluorescence (**C**,**D**). (**A**) Diffuse mesangial and segmental endothelial hypercellularity with segmental necrosis and glomerular neutrophil infiltration (haematoxylin and eosin, 400×). (**B**) A cellular crescent (periodic acid-silver methenamine, 200×). (**C**), C3 (++) deposition in mesangial areas and capillary loops. (**D**) No IgA deposition.

**Figure 2 diagnostics-12-00772-f002:**
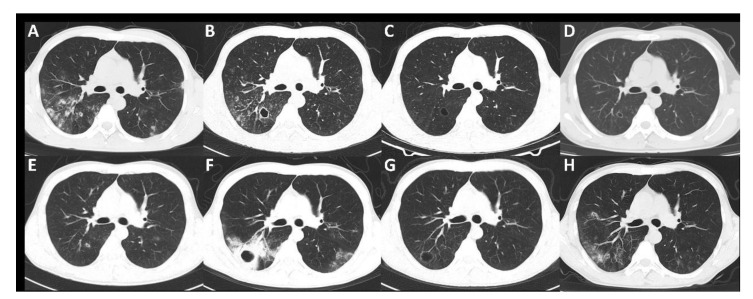
Changes in chest CT imaging during treatment. (**A**) In October 2018, before anti-tuberculosis treatment, patchy shadows were seen in the posterior segment of the right upper lobe and the dorsal segment of the left lower lobe. (**B**) In December 2018, after 3 months of anti-tuberculosis treatment. A new cavity appeared in the lower lobe of the right lung. (**C**) In February 2019, after receiving levofloxacin, glucocorticoids and cyclophosphamide for one month, the multiple patchy shadows in both lungs were alleviated, and the right pulmonary cavity was reduced, with a thinner wall. (**D**) In April 2019, after receiving glucocorticoids and cyclophosphamide for 3 months, there were more patchy shadows in both lungs, and the right pulmonary cavity was smaller, but the cavity wall was thicker. (**E**) In May 2019, after receiving ceftazidime treatment, the right lung cavity shrank. (**F**) In November 2019, after receiving glucocorticoid and azathioprine therapy for 4 months, new-onset patchy consolidation in both lungs was observed. A thick-walled cavity appeared in the right lower lung. (**G**) In November 2019, after receiving meropenem, cefepime and moxifloxacin treatment, the lesions in both lungs were significantly reduced, and the cavity wall of the right lower lobe became thinner. (**H**), In March 2021, after receiving prednisone, azathioprine and mycophenolate mofetil, multiple patches and nodular shadows in both lungs, some of which showed cavitation, were observed.

**Figure 3 diagnostics-12-00772-f003:**
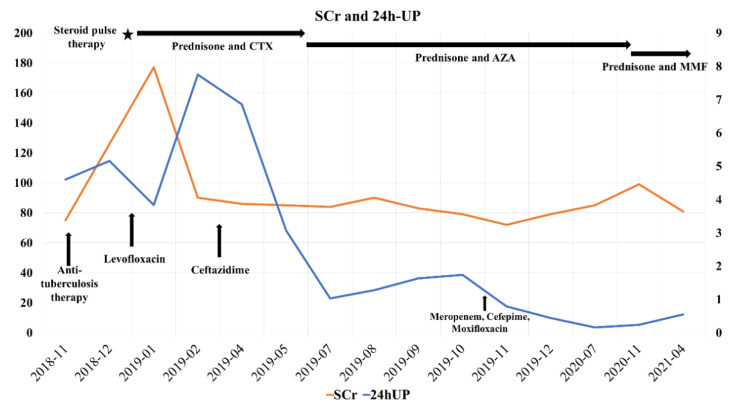
Changes in serum creatinine (orange line, µmol/L, left axis), 24 h urine protein (blue line, g/24 h, right axis) and the treatment of the patient. Immunosuppressive therapy: In January 2019, the patient received steroid pulse therapy (methylprednisolone 1 g every day for 3 days). From January to June 2019, the patient was treated with prednisone (60 mg/d for 7 weeks, subsequently reduced by 5 mg every two weeks to 10 mg/d maintenance) and cyclophosphamide (0.2 g every other day, cumulative dose 16 g) orally. From July 2019 to December 2020, the patient received oral prednisone (10 mg/d) and azathioprine (50 mg/d for 6 months, subsequently 75 mg/d for 1 year). From January to March 2021, the patient received oral prednisone (30 mg/d for 1 month, 50 mg/d for 1 month, subsequently reduced by 5 mg every week) and mycophenolate mofetil (1.5 g/d for 2 months). Antibiotic therapy: From October (Figure 2A) to December 2018 (Figure 2B), the patient received anti-tuberculosis treatment (isoniazid 0.5 g/d, pyrazinamide 0.75 g/d and ethambutol 1.5 g/d). In January, April (Figure 2D) and November (Figure 2F,G) 2019, the patient received short-term antibacterial treatment three times. ★ means steroid pulse therapy.

**Figure 4 diagnostics-12-00772-f004:**
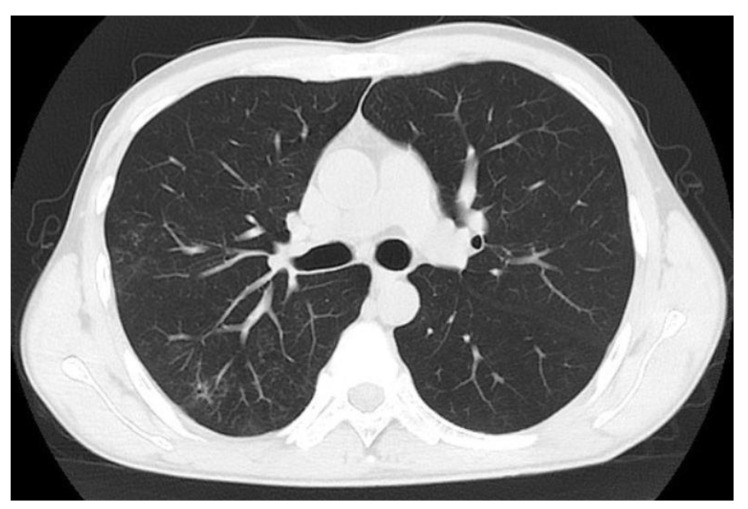
June 2021, after 2 months of treatment with levofloxacin, inhaled amikacin sulfate and low-dose prednisone (10 mg/d). Chest CT scan showed that multiple patches and cavities in the lung were obviously absorbed.

## Data Availability

Not applicable.

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
