# Peer review of "Vasculitis Secondary to Pulmonary Bacterial Infection: A Case Report"

_diagnostics, 2022, doi:10.3390/diagnostics12040772_

Round 1
Reviewer 1 Report
This case report on vasculitis secondary to pulmonary infection is an exciting piece of work. The manuscript is well written, and this report explicitly describes experimental findings. I would support the publication of this report after feedback on a short comment. Do authors suggest including anti-inflammatory drugs to control high inflammation from killed bacteria after antibiotic therapy?
Author Response
Point 1: Do authors suggest including anti-inflammatory drugs to control high inflammation from killed bacteria after antibiotic therapy?
Response 1: Thank you for the question. Yes,we suggest anti-inflammatory drugs to control high inflammation from killed bacteria after antibiotic therapy in our patient. As mentioned in the Case Presentation (Page 5, Paragraph 2) and Discussion section (Page 6, Paragraph 6), given that steroids may attenuate the systemic inflammatory response, we continued to maintain glucocorticoids at low doses (oral prednisone 10 mg/d). The treatment of vasculitis caused by bacteria mainly depends on antimicrobial agents, and a short course glucocorticoid and/or immunosuppressive agents may sometimes be necessary. Most postinfectious glomerulonephritis self-heals after the infection is controlled. However, if there is a combination of necrosis and crescents, steroid therapy is often needed. If the number of crescents is large as our patient, combined immunosuppressive therapy is often necessary.
Reviewer 2 Report
Very interesting case report but the first diagnosis is unclear. The whole case report is based on the diagnosis of bacterial pneumonia but there is no documentation for the first episode. How can we be sure of that diagnosis ? it seems that the infection with Pseudomonas aeruginosa or Staphylococcus occurs later, but there's no evidence for the initial episode.
Author Response
Point 1: The whole case report is based on the diagnosis of bacterial pneumonia but there is no documentation for the first episode. How can we be sure of that diagnosis ?
Response 1: Limited to the number of words in the article, we did not introduce in detail the diagnosis and treatment process of the patient in the local hospital at the time of onset. In November 2017, January 2018 and March 2018, the patient admitted to a local hospital 3 times due to fever, cough and expectoration, and his chest CT showed patchy shadows in both lungs. Antibiotic treatment was effective, so we consider that the patient initially pneumonia diagnosis was started from Novermber 2017. However, when the patient’s symptoms recurred in July 2018, antibiotic as well as anti tuberculosis treatment were not effective. So the patient was hospitalized in our hospital for the first time in December 2018 and considered the diagnosis of ANCA negative vasculitis.
We added specific information about the onset of the disease in the Case Presentation section (Page 2, Paragraph 1) to clarify. Thank you for your comments.
Point 2: It seems that the infection with Pseudomonas aeruginosa or Staphylococcus occurs later, but there's no evidence for the initial episode.
Response 2: As you mentioned, the initial causative agent of the pneumonia was not clear since the patient was not examined for respiratory pathogens during treatment in the local hospital. But since the patient visited our hospital in 2018, he underwent 4 times of bronchoalveolar lavage fluid (BALF) cultures, revealing persistent positive Pseudomonas aeruginosa and positive benzoxicillin-sensitive Staphylococcus aureus twice. Since the lesions of the patient’s lungs are similar among different visits, we believe the pathogens are reliable based on the consistant results.